# NCI–Clinical Trial Accrual in a Community Network Affiliated with a Designated Cancer Center

**DOI:** 10.3390/jcm9061970

**Published:** 2020-06-24

**Authors:** Daniel J. Kim, Dan Otap, Nora Ruel, Naveen Gupta, Naveed Khan, Tanya Dorff

**Affiliations:** City of Hope Comprehensive Cancer Center, Department of Medical Oncology and Developmental Therapeutics, Duarte, CA 91010, USA; danieljkim@coh.org (D.J.K.); dotap@coh.org (D.O.); nruel@coh.org (N.R.); ngupta@coh.org (N.G.); nakhan@coh.org (N.K.)

**Keywords:** clinical trials, community practice, minorities, ethnicity, race

## Abstract

Most cancer care is delivered in the community, while most clinical trials exist in academic centers. We analyzed clinical trial accrual of a tertiary care cancer center and its affiliated community sites to better understand what types of trials accrued at the community sites and whether community accrual increased ethnic diversity. The institutional clinical trial database was searched for solid tumor accruals during 2018–2019. Patient’s race was abstracted, and trial’s funding source, phase, and disease type/stage were tabulated. Of 3689 accruals, 133 were at community sites, representing 26 unique trials while the main campus accrued to 93 unique trials. Community site accruals were highest for breast and colorectal cancer, but patients with less common cancers such as renal, nasopharyngeal, and gastric cancer were also accrued at community sites. Accruals occurred to randomized trials, as well as phase Ib and translational biomarker studies. Minority patients constituted 20.0% and 32.5% of community site accruals for therapeutic and non-therapeutic trials respectively, compared to 20.6% and 29.8% of main campus accruals for therapeutic and non-therapeutic trials, respectively. We conclude that community sites affiliated with an academic cancer center can accrue to a broad spectrum of clinical trials while enhancing racial diversity in participation of clinical trials. Further expansion of access to clinical trials in community sites is necessary to broaden patient access to state-of-the-art and next-generation treatment options.

## 1. Introduction

Community oncology practices provide approximately 55% of all care for cancer patients in the United States [1]. Providing access to clinical trials in community oncology practices has been a major initiative of the National Cancer Institute (NCI, Rockville, MD, USA), as evidenced by the creation of the National Community Oncology Research Program (NCORP, Bethesda, MD, USA) in 2014. The NCORP was created by consolidating the previous NCI supported networks of the CCOP–community clinical oncology program and the NCI Community Cancer Centers Program (NCCCP, Bethesda, MD, USA). Up to a third of clinical trial accrual in NCI studies is contributed by these community oncology practices [2], yet the overall clinical trial accrual remains lower than desirable: fewer than 5% of adult cancer patients enroll in clinical trials [3], despite 70% of cancer patients expressing interest in enrolling in a clinical trial [4].

Aside from increasing treatment options and providing access to the latest innovations in cancer care, participation in clinical trials has been shown to improve overall treatment quality vis-à-vis utilization of optimal standards of care [5]. One barrier for community practice physicians has been cost; clinical trial enrollment takes time, which takes resources away from overall clinic flow, and research staff are required regardless of how robust accrual is. Understanding the types of clinical trials that readily accrue in the community setting could help improve trial selection to foster financial sustainability, since fewer trials, each of which accrue larger numbers of patients, is more efficient in the setting of having limited clinical research staff at community sites. 

In addition, inadequate representation of racially diverse cancer patients has been noted to be a major problem in the applicability of most clinical trials to everyday practice. For example, clinical trials in metastatic castration-resistant prostate cancer (CRPC) were found to have 80% under-representation of African American patients [6]. In the case of geriatric patients, a study of NCI-sponsored trials found that patients over 65 made up only 32% of participants, whereas they made up 61% of patients affected by the cancers [7]. Given that community oncology practices may be located closer to patients’ homes, the availability of trials in a local setting could reduce potential barriers for patients to access clinical trials. Furthermore, if community practices are located near specific ethnic enclaves, their participation in clinical trials could help alleviate the lack of racial diversity, as indicated by the NCCCP initiative [8].

City of Hope is an NCI-designated comprehensive cancer center in Duarte, California, with an extensive network of 21 community practice sites located within its vicinity in Southern California, thus allowing for delivery of specialized care near where patients reside. As part of the overall mission of City of Hope, select clinical trials have been opened to accrual at six of the community sites which service a high enough patient volume to support research recruitment. We sought to characterize and evaluate clinical trials accrual with a focus on City of Hope’s community sites, and whether racial diversity was enhanced in patients accrued at community sites compared to the main cancer center campus.

## 2. Material and Methods

After obtaining IRB exemption for the study, we identified the individual accrual events in our clinical trials database during the calendar years 2018 and 2019 at all City of Hope sites. This dataset was extracted from our institution’s secure online trial database, devoid of any personal identifiers. Inclusion criteria was prospective adult clinical trials. Study title, sponsor, location of accrual, and date of study were extracted from the dataset. We decided ahead of time to exclude trials which were felt to reflect highly tertiary patient populations, such as hematologic malignancy trials which, at City of Hope, were likely to be heavily transplant and chimeric antigen receptor T cell therapy-based, or any other trials felt to introduce bias into the analysis.

Results were tabulated for comparison between main campus and community sites and stratified by the type of trials. The Chi-square and Fisher’s exact tests were used to test the differences in distribution of race, minority status, and age groups between community sites and main campus participants.

## 3. Results

Of the 33,558 individual accrual events identified in our analysis, 23,809 accrual events were identified to be retrospective in nature and thus excluded, along with an additional 5625 accrual events which were identified to be trials involving hematologic malignancies. Sixteen pediatric trial accrual events were also excluded, as well as 419 healthy volunteer accruals. Also excluded were two high-accruing non-therapeutic trials specifically recruiting only Hispanic women, as ethnicity was not a variable and thus biasing the minority accrual question. 

Our final analysis was thus based on 3689 accrual events. Community sites in the City of Hope network accrued to 26 distinct, solid tumor clinical trials (*n* = 22 therapeutic, *n* = 4 non-therapeutic) during the years 2018–2019, compared with accrual to 93 solid tumor trials at the main campus during the same time period. A description of all therapeutic trials which accrued at least one subject at a community site is delineated in Table 1. During the same time period, an additional 57 trials were open at community sites but did not accrue patients there.

Overall, 133 subjects were accrued at 6 community sites (*n* = 58 therapeutic) and 3556 at the main campus. These included randomized phase II or III trials (*n* = 13), as well as (*n* = 7) phase II trials, and (*n* = 2) phase Ib trials. Half of the trials accrued to in the community sites (*n* = 13) were cooperative group trials, four were industry, one was NCI/CTEP sponsored, and seven were investigator-initiated trials commonly called investigator-sponsored trials (IST). Trials which did not register accrual at community sites were primarily phase II (30/57; 52.6%) and phase III (21/57; 36.8%), with only five phase I trials open and not accruing (8.8%).

Most trials that registered community accruals were for breast cancers (*n* = 8), followed by gastrointestinal (6), genitourinary (3), non-small cell lung cancer (3), melanoma (1), and all solid tumors (1). Highest accruing trials in the community sites included a phase III industry-sponsored trial in metastatic refractory colorectal cancer, which accrued more patients in the community than at the main cancer center (*n* = 10 compared to *n* = 4 respectively). The other top accruing studies (*n* = 6 in community) were for breast cancer patients, one for prevention and one for first-line treatment of metastatic disease. Studies which were open in the community, but did not accrue, were distributed across tumor types; 13 gastrointestinal (22.8%), breast and lung (each 19.3%), melanoma/sarcoma and genitourinary (each 14.0%), head and neck and gynecologic (each 5.3%).

The ethnic composition of accruals in the community sites and the main campus is summarized in Table 2. Minority accrual ranged from 20.6% to 21.8% at the main campus and 15.6% to 20.0% at community sites, although there was a higher rate of accrual of African American subjects to non-therapeutic trials at community sites. Distribution of patient age was different between accrual sites (Table 3) within therapeutic and non-therapeutic studies (*p* < 0.01), as we saw a larger proportion of older patients at community sites than on main campus. Among adult subjects, 22/58 (37.9%) subjects were > 70 years old in the community, compared to 498/1828 (27.3%) patients accrued in Duarte. In contrast, 238 (13.0%) adult subjects on main campus vs. zero patients at community sites were under the age of 40.

A complete list of trials accrued to at each site can be found in Appendix A. Notably, trials accrued to at community sites included the full spectrum of disease, from cancer prevention to adjuvant and neoadjuvant trials, as well as trials for metastatic and refractory cancer patients. Non-interventional trials which accrued at community sites included biomarker studies, both for molecular characterization as well as for predicting toxicity, and geriatric oncology trials. Studies not open in the community included supportive care trials such as a trial studying bright light to reduce frailty during androgen deprivation therapy for prostate cancer patients and studies of the perception of genomic profiling and of immunotherapy expected outcomes. Most phase I trials were not open in the community (26/32).

## 4. Discussions

Community sites associated with an NCI-designated cancer center may represent a unique opportunity to expand clinical trial access into underserved and under-represented populations, given the proximity of the sites to serve their respective local communities and the ability to leverage research staff and resources from the main campus. A review of more than 33,000 individual accrual events at City of Hope revealed that affiliated community sites accrued patients to a wide selection of solid tumor trials of various cancer histology, stage of disease, and therapeutic/non-therapeutic aims. Whereas it was anticipated most trials that successfully accrued patients at community sites would be later phase, cooperative group trials, there were also a relatively high accrual of investigator-sponsored phase II, and even phase Ib, trials. Similarly, it was anticipated that the most common malignancies would make up the largest percentage of accruals in the community sites, but uncommon tumor types such as hepatocellular, kidney, and pancreatic cancers also accrued at community sites.

City of Hope community practice sites are strategically located throughout five counties across Southern California (Ventura, Los Angeles, San Bernardino, Riverside, and Orange), aiming to provide cancer care to all communities, including socioeconomically disadvantaged populations and communities with cancer disparities. Our analysis did reveal a higher accrual of African American patients to non-therapeutic trials at the community sites compared to main campus but did not significantly improve diversity of trial accrual as had been hoped. Loree et al. [9] had reported African Americans and Hispanics representing 3.1% and 6.1% of trial participants respectively, and City of Hope’s rate of African American representation in trial accrual was similarly low. A limitation of our analysis is that our percentages are derived from accrued subjects rather than all subjects; that is to say, the denominator are trial participants, rather than all screened patients or all eligible patients or all patients seen at the clinic, as such data was not available for analysis. Our assessment was also limited by a relatively high rate of “unknown” ethnicity among community accruals (22.4%); this can be due to incomplete datasets or patients declining to answer the question. In order to minimize missing data, standard operating procedures for research staff and centralization of clinical data management could improve consistency and data quality.

Community sites did enroll a higher rate of geriatric patients, which may reflect the difficulty elderly patients face in traveling greater distances to a comprehensive cancer center in order to obtain oncology care. Admittedly, the City of Hope experience may be unique, as geriatric oncology is a flagship program with dedicated resources. However, this finding is an encouraging signal, and warrants further study as increasing accrual of geriatric patients to cancer protocols is an important goal for the oncology community given that many cancers have substantially higher prevalence in people over the age of 65.

Additional limitation of our study involves the inherent selection process involved in extending clinical trial offerings to the community sites. Based on limited resources and anticipated accrual rates, only a select few clinical trials were extended from the main campus to the community sites, and of the 21 community sites only six higher-volume sites were selected for clinical trial participation. Trial accrual rate could have been higher for the community sites if trial offerings were identical across main campus and community sites, but such an offering would require far greater investment into trial resources at all of the community sites and may not be practical or feasible. Our analysis also could not capture the number of patients who were referred from the community sites to the main campus for enrollment into trials not offered at the community sites, thus shifting trial accrual numbers from community sites to main campus. We did notice, however, that a bladder cancer trial enrolled subjects at the main campus and then transitioned ongoing cycles of treatment to community sites due to patient travel burden (Table 1). This flexibility is an example of how academic-community collaboration can optimize both accrual and patient-centered care and could represent an efficient “just in time” model of opening the right trials in the right areas. We concur with the value of main campus regularly meeting with community sites to review currently available clinical trials [10], and we additionally propose that the community sites must be actively engaged in and collaborate with the main campus in selecting the trials offered at the community sites based on needs assessment of their unique patient population and community site clinician interests.

In contrast to published experiences, however, we found that phase I and even non-therapeutic trials did not present an insurmountable barrier to accrual in community oncology practices. In a previous survey of 51 community practices, which included both federally sponsored and private non-academic affiliated practices, half of the practices self-reported that they did not accrue to phase I trials, 53% did not accrue to investigator-initiated trials, and 33% did not enroll on correlative science trials [11]. It is likely that many investigator-initiated trials and translational “non-therapeutic” trials can meet the needs of community oncology patients, and that academic-affiliated community sites are best poised to engage patients in accrual to these types of studies.

## Figures and Tables

**Table 1 jcm-09-01970-t001:** Characteristics of therapeutic clinical trial which accrued subjects at City of Hope community practice sites from 2018–2019.

Disease	Stage of Disease	Phase of Trial	Sponsor	Community Site Accrual	Main Campus Accrual
Breast Cancer	Prevention	II	IST	6	7
Breast Cancer	Adjuvant	III, randomized	Cooperative group	2	2
Breast Cancer	Adjuvant	III, randomized	Cooperative group	3	0
Breast Cancer	Metastatic 1st line	II	IST	4	8
Breast Cancer	Metastatic, over 60, Her2+	II	IST	2	11
Breast Cancer	Survivorship	II, randomized	Cooperative group	3	20
Colorectal Cancer	Metastatic, previously treated	III, randomized	Industry	10	4
Colorectal Cancer	Metastatic RAS-mutated, previously treated	I	IST	3	18
Colorectal Cancer	Metastatic previously treated	III	Industry	5	2
Colorectal Cancer	Metastatic previously treated	Ib	Industry	1	0
Gastric/GEJ	Metastatic, previously treated	I/II	NCI	2	2
Hepatocellular carcinoma	Advanced/Metastatic 1st line	III, randomized	Industry	2	11
Kidney Cancer	Adjuvant	III, randomized	Industry	3	9
Nasopharyngeal	Primary	II/III, randomized	Cooperative group	1	4
NSCLC	Adjuvant	III, randomized	Cooperative Group	1	4
NSCLC	Adjuvant, EGFR mutant	III, randomized	Cooperative group	1	4
NSCLC	Metastatic	I/II	IST	2	3
NSCLC	Metastatic(oligo) add SBRT	II/III, randomized	Cooperative Group	1	2
NSCLC	Metastatic, first-line, EGFR mutant	IV (elderly)	Industry	1	0
Pancreas	Advanced/Metastatic	Pilot	IST	2	4
Prostate Cancer	mCRPC	II randomized	IST	1	4
Urothelial	Neoadjuvant	II	Cooperative Group	2 *	14

* Accrued at main campus, transitioned during study treatment to community site. GEJ = gastro-esophageal junction. IST = Investigator Sponsored Trials. NSCLC = non-small cell lung cancer.

**Table 2 jcm-09-01970-t002:** Accrual at sites, stratified by race.

**Therapeutic**			
**Race**	**Main Campus Therapeutic *n* = 1832**	**Community Site Therapeutic *n* = 58**	***p*-Value ***
	**Number (%)**		
African American	76 (4.1)	3 (5.2)	0.4
American Indian or Alaska native	4 (0.2)	0	
Asian	255 (13.9)	5 (8.6)	
Native Hawaiian or Pacific Islander	7 (0.4)	1 (1.7)	
Non-White Multiracial	2 (0.1)	0	
White Multiracial	12 (0.7)	0	
White	1382 (75.4)	36 (62)	
Unknown	94 (5.1)	13 (22.4)	
%minority (non-White/total)	20.6%	20.0%	1.0
**Non-Therapeutic**			
**Race**	**Main Campus Non-Therapeutic *n* = 1192**	**Community Site Non-Therapeutic *n* = 40**	
	**Number (%)**		***p*-Value**
African American	64 (5.3)	5 (12.5)	0.04
American Indian or Alaska native	5 (0.4)	0	
Asian	157 (13.2)	0	
Native Hawaiian or Pacific Islander	3 (0.3)	0	
Non-White Multiracial	0	0	
White Multiracial	4 (0.3)	0	
White	837 (70.2)	27 (67.5)	
Unknown	122 (10.2)	8 (20)	
%minority (non-White/total)	29.8%	32.5%	0.5

* *p* values calculated for distribution of known race or ethnicity, omitting unknowns from calculations.

**Table 3 jcm-09-01970-t003:** Age distribution of subjects accrued to clinical trials at the main campus and at community sites.

	Therapeutic Trials		Non-Therapeutic Trials	
Age Group	Main Campus	Community Sites	Main Campus	Community Sites
**19–39**	238 (13.0%)	0 (0.0%)	154 (13.4%)	1 (2.5%)
**40–59**	563 (30.8%)	21 (36.2%)	303 (26.4%)	4 (10.0%)
**60–69**	528 (28.9%)	15 (25.9%)	292 (25.5%)	15 (37.5%)
**70+**	498 (27.3%)	22 (37.9%)	398 (34.7%)	20 (50.0%)
**Unknown**	0 (0.0%)	0 (0.0%)	0 (0.0%)	0 (0.0%)

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
