# Peer review of "NCI–Clinical Trial Accrual in a Community Network Affiliated with a Designated Cancer Center"

_jcm, 2020, doi:10.3390/jcm9061970_

Round 1

Reviewer 1 Report

Kim et al. explore the rate of accrual to clinical trials in community sites associated with an NCI-designated cancer center. Using 3,689 individual trial accruals (133 accrued at six community sites and 3,556 accrued at the cancer center), they evaluate which types of clinical trials were successful at accrual and whether ethnicity influences accrual rates as compared to the main campus. They conclude that community sites were successful at accruing patients to a wide variety of solid tumor trials, and that there was no difference in minority accrual at the community sites.

 Major comments:

  1. The setting is unclear for the general reader. Please indicate where is COH located, how many community sites are associated with the cancer sites and how many of them actually accrued patients? If only a selected number of sites participates (6 out of 20?), then please explain why.
  2. The first aim of the study was to evaluate which types of clinical trials were most successful at accrual by community sites (lines 54-55). The paper does not provide a clear answer to this question: rather, the types of trials accrued to is summed up (lines 91-94), but the proportion of accruals for separate types of trials is not compared to those accrued at the main campus. Is there a difference between the main campus  and the sites with respect to  the number of patients who were initially contacted to participate in a trial and who actually enrolled in the trial?  Such a comparison is necessary to reveal exactly which types of clinical trials accrue comparably to the main campus and which are over- or underperforming.  Supplemental Table S1 (ST1 ) summarizes all trials in which the community sites participated.  Table 1 provides the accrual numbers for a selection these  trials . The community hospitals did not accrue patients  to the trials not listed in table 1? The authors conclude that community practice sites were successful at accruing to a broad spectrum of clinical trials (Abstract, lines 18-19 and Discussion, lines 108-110), but did not specify a reference frame to define “successful” in this sense. Intuitively, 133 accruals for 55% of cancer patients in community centers (as stated in line 25) as opposed to 3,556 for the remaining 45% of cancer patients in the main campus seems exceedingly disproportionate. The authors need to clearly define what is defined as “successful” accrual rates.
  3. The second aim of the study was to evaluate whether ethnic diversity was enhanced in patients accrued at community sites compared to the main campus (lines 55-56). Their answer is not clear. In the Abstract, the authors imply (without statistical substantiation) that accrual rates may differ based on cancer type (lines 16-18). Yet in the Discussion, they conclude that there was no difference between community sites and main campus (lines 129-130). This confusion may be caused by the fact that the authors unequally divided the main campus into subgroups. The ethnic diversity of the 87 and 46 patients recruited by the community sites are compared to the two largest accruing disease teams (breast and genitourinary) at the cancer center. This potentially induces an unnecessary risk of bias. Either include all accruals regardless of cancer type or be equally selective for main campus and community sites. Statistical analyses must be performed to substantiate a claim on recruitment efficacy and when numbers cannot be compared due to lack of power, then this should be acknowledged.
  4. The introduction features a comment on the underrepresentation of elderly patients in clinical trials identified by Lewis et al. (J Clin Oncol, 2003) and hypothesize that this may improve as a result of accrual in community sites due to the reduction of travel barriers,). Another reason for underrepresentation not mentioned in the paper may be the trials’ inclusion criteria, which often exclude frail patients with comorbidities. The impact of the study would increase if an analysis comparing representation of elderly patients between community hospitals and the main campus would indeed reveal a higher proportion of elderly patients accrued to clinical trials.

Minor comments:

  1. The manuscript focuses equally on both overall accrual rates and ethnic composition of the accruals and considers these the main outcomes of the study in the Introduction (lines 54-56). However, the focus on ethnicity is not reflected by the aim as stated in abstract (lines 9-10). The abstract should be adjusted to reflect the overall aim of the study.
  2. In lines 39-41 (“Understanding the types … relatively high accrual”), the authors seem to suggest that trials with high potential accrual should be avoided by community practices, whereas in the remainder of the manuscript they imply that a higher accrual in community practices is desirable. This sentence should be rewritten to either correct or explain this contradiction.
  3. The Materials and Methods does not sufficiently clarify how the study was conducted. The first paragraph primarily contains the n amounts of accruals (lines 63-66), which could only have been determined during or after conduct of the study and should thus be reserved for the Results section. The second paragraph contains a long sentence (lines 67-69) describing communication methods, of which the purpose is unclear as it is not touched upon in the rest of the paper. The authors should revise this section to provide a clear image of how they conducted the study. In addition, the authors need to specify which accrual-specific variables were extracted from the clinical trials database (age, sex, diagnosis, ethnicity, others?).
  4. Lines 77-79 (“These included randomized … non-interventional studies”) are misplaced, as they seem to refer to the 26 clinical trials that community practices accrued to (lines 74-75) rather than the overall amount of accrued subjects (lines 76-77).
  5. The tables include multiple abbreviations that are not explained in the table nor in the main text. 
  6. The manuscript contains small textual errors. Examples include “introduction” not capitalized (line 24), missing plural forms on “adult cancer patients” (line 32) and “all solid tumors” (line 82), a capitalized N (line 84) and “IST” not fully capitalized in all instances in Supplemental Table 1.

Author Response

please see the uploaded document for our point-by-point response to reviewer 1 comments

Reviewer 2 Report

The authors provide results of a data analysis based on a local clinical trial registry at City of Hope, U.S.A.

Their main aim was a characterisation of trials comparing recruitment at community centers vs. main cancer center campus.

The analysis if based on the comparison on the trial-level and patient-level data for ethnicity. The methodology to compile the final data set is described sufficiently.

The manuscript is very descriptive and statistical analysis results in absolute and relative frequencies. Statistical analysis of the mainly descriptive comparisons could be improved to support the main research question.
Especially, statistical test could be used to assess differences between community and main cancer center.

Eg. Table 2 shows more or less a contingency table between ethnicity and type of center combined with trial intention. A statistical test might be helpful to assess the observed differences (relative frequencies) to "prove" a different distribution of ethnicities between the main factor (community vs main center). Other comparison also might be informative: e.g. therapeutic vs. non-therapeutic?

A similar analysis could support a variation in study stages (I,II,III) and other study characteristics.

I wonder why other baseline patient characteristics were not included into the analysis. E.g. differences in age, patient performance status, grading of the disease (worse patients might preferably visit the main campus cancer center?) , and others might be available and also very informative for a comparison.

Also an analysis on the overall recruitment rates would be very informative.
The main question would be: "Does the inclusion of community centers speed up accrual?" The reported accrual numbers (e.g. Table 1) could be set into relation to the planned accrual (if available). The conclusions (see discussion), to foster community centers for recruitment might be set into context with these numbers. Initiation of a new trial site (maybe community-based) when expected recruitment is low probably does not justify the effort.

Given the narrow scope of new information in the results (trial stage, diseases, ethnicity) I would suggest to change the manuscript's title. Currently, the title suggests a paper that gives a wide range of information on various aspects of accrual in community sites - which is not given in the current version.

Overall, I have the feeling that the paper has a certain value and provides new information to readers. If it possible to add statistical test results for the main questions, and potentially extend the analysis to a broader set of patient-level characteristics, the paper would benefit greatly.
A modification of the title should be considered, to better reflect the scope of analysis and results.

Author Response

please see the uploaded document for point-by-point response

Round 2

Reviewer 1 Report

The authors have adequately addressed the comments, but not the issue regarding the age distribution of patients in the community sites and main campus. A  chi-square test revealed a significantly (?) older patient population at the community sites. The full breakdown of the age bins is not provided. Was 70-79 compared to <70? The value given for Chi2 seems odd (shoudl be 1.6) or that its P value. The P value when using the Fisher's exact test for this 2x2 comparission is likely not significant given the small number of patients (58) at the community sites. When a significant increase cannot be demonstrated, then "significant" in line 108 should be avoided. 

Author Response

Thank you for your further input. we have addressed the comments and uploaded a revised manuscript.

The authors have adequately addressed the comments, but not the issue regarding the age distribution of patients in the community sites and main campus. A  chi-square test revealed a significantly (?) older patient population at the community sites. The full breakdown of the age bins is not provided. Was 70-79 compared to <70? The value given for Chi2 seems odd (shoudl be 1.6) or that its P value.

we have added Table 3 which shows the complete breakdown of ages and have added additional detail in the manuscript.

The P value when using the Fisher's exact test for this 2x2 comparission is likely not significant given the small number of patients (58) at the community sites. When a significant increase cannot be demonstrated, then "significant" in line 108 should be avoided. 

Agree and apologize for this misuse of the word. we have removed the word significant from that line.

Reviewer 2 Report

Thank you for taking my comments into account.

I understand your wish to leave this work on a more descriptive level and support acceptance.

Author Response

Thank you for your input.